# *Pseudomonas Aeruginosa* Induced Cell Death in Acute Lung Injury and Acute Respiratory Distress Syndrome

**DOI:** 10.3390/ijms21155356

**Published:** 2020-07-28

**Authors:** Rushikesh Deshpande, Chunbin Zou

**Affiliations:** 1Department of Environmental and Occupational Health, Graduate School of Public Health, University of Pittsburgh, Pittsburgh, PA 15213, USA; rhd10@pitt.edu; 2Division of Pulmonary, Allergy and Critical Care Medicine, University of Pittsburgh, Pittsburgh, PA 15213, USA

**Keywords:** *Pseudomonas aeruginosa*, cell death, apoptosis, ferroptosis, acute lung injury/acute respiratory distress syndrome, lung infection

## Abstract

*Pseudomonas aeruginosa* is an important opportunistic pathogen responsible for the cause of acute lung injury and acute respiratory distress syndrome. *P. aeruginosa* isthe leading species isolated from patients with nosocomial infection and is detected in almost all the patients with long term ventilation in critical care units. *P. aeruginosa* infection is also the leading cause of deleterious chronic lung infections in patients suffering from cystic fibrosis as well as the major reason for morbidity in people with chronic obstructive pulmonary disease. *P. aeruginosa* infections are linked to diseases with high mortality rates and are challenging for treatment, for which no effective remedies have been developed. Massive lung epithelial cell death is a hallmark of severe acute lung injury and acute respiratory distress syndrome caused by *P. aeruginosa* infection. Lung epithelial cell death poses serious challenges to air barrier and structural integrity that may lead to edema, cytokine secretion, inflammatory infiltration, and hypoxia. Here we review different types of cell death caused by *P. aeruginosa* serving as a starting point for the diseases it is responsible for causing. We also review the different mechanisms of cell death and potential therapeutics in countering the serious challenges presented by this deadly bacterium.

## 1. Introduction

Acute lung injury and acute respiratory distress syndrome (ALI/ARDS) are a severe public concern worldwide. Pneumonia from viral and bacterial infection remains a major cause of infectious death. The current pandemic of COVID-19, a viral infection caused by severe acute respiratory syndrome coronavirus 2 (SARS-CoV2), mainly causes ALI/ARDS and affects almost all countries worldwide. More than 10 million patients have been diagnosed with COVID-19 with half of million deaths, and the numbers of confirmed cases and deaths are continuously increasing [1]. Viral infection destroys host defense with exhausted immune response, that is easily followed by a secondary bacterial infection. The secondary bacterial infection always renders poor prognosis of the patients. Among the microbial pathogens involving secondary infection, *Pseudomonas aeruginosa*, a Gram-negative rod-shaped bacterium is the leading opportunistic pathogen isolated from patients with nosocomial infection. *P. aeruginosa* is present in the respiratory tract of many of the patients with long-term ventilation more than 7 days [2]. It is also the leading cause of deleterious chronic lung infections in patients suffering from cystic fibrosis (CF) [3]. *P. aeruginosa* infection is the major reason for morbidity in people with chronic obstructive pulmonary disease (COPD) as well [4]. Patients with viral infection, trauma, and cancer leads to host immunosuppression that largely enhances secondary infection with *P. aeruginosa*. *P. aeruginosa* is in most of the cases multi-drug resistant and no effective cure has been developed. Diseases linked with *P. aeruginosa* infection display high mortality rates and are particularly challenging for treatment in patients with compromised immunity [2].

*P. aeruginosa* is able to adapt to a specific functional role to escape the host defense using highly complex intracellular and intercellular signaling networks. Besides, *P. aeruginosa* releases a large number of virulence effectors [5,6]. This adaptive ability however has variances according to the types of clones and strains in the *P. aeruginosa* population worldwide [7].The Pathogenicity of *P. aeruginosa* is a result of long-time adaption and evolution. Pathological changes include inflammatory response, cytokine secretion, immune cell infiltration, air-blood barrier dysfunction, leakiness, and cell death.

Once *P. aeruginosa* enters into the airway tract, the bacteria replicate and colonize in the lumen of the airway and alveolar sacs. Part of the bacteria will be engulfed and cleared by lung residential macrophages. Meanwhile, *P. aeruginosa* activated macrophages release inflammatory factors, result in immune cell infiltration, and subsequent hyper-reacted cytokine storm may damage the respiratory system. On the other side, part of the bacteria will invade into lung epithelial cells through the damaged physical protection barrier of mucin or surfactant films. The bacteria adhere on the surface of epithelial cell membrane and penetrate the membrane to replicate and colonize in the cells [8,9,10,11,12]. The flagella and Type IV pili (TFP) play the role of adhesins for host cells [9]. Other adhesins such as cup fimbrial adhesins [13], lectins PA-IL (LecA) and PA-IIL (LecB) [14] have also been identified. In the study by Hayashi N. et al. [15], it was observed that the bacteria are associated with Caco-2 cells even in the absence of intact pili, suggesting that *P. aeruginosa* can bind to Caco-2 cells using other adhesion systems. Firstly, the replication and colonization of the invaded bacteria may physically damage the integrity of the lung epithelial cells. Secondly, the bacteria cause host inflammatory response via multiple channels of molecular mechanisms that induce lung epithelial cell death. Furthermore, *P. aeruginosa* may directly release a number of toxins that directly destroy cellular membranes to lead to cell death. In addition, the dead cells and debris may release toxic cellular components that may further augment inflammation and autoimmune responses. *P. aeruginosa* infection mediated lung epithelial cell death destroys the integrity of the air–blood barrier, and leads to leaks and edema, that will attract peripheral circulating neutrophils, microphages, and lymphocyte infiltration and enhance inflammatory response. Therefore, *P. aeruginosa* mediated lung epithelial cell death is a major mechanism in the pathogenesis of ALI/ARDS. Our understanding on the underlying molecular mechanisms of *P. aeruginosa* induced lung epithelial cell death is yet to be studied for the development of an effective therapy against the pathogen and the related illnesses.

## 2. Distinct Cell Death Mechanisms

Cell death is a phenomenon exclusively existing in multicellular organisms, both in physiological processes and in responding to pathological stimulus. In order to maintain homeostasis of tissues as well as eliminate potentially harmful stimulus, cell death is a crucial process [16]. Mounting studies have dissected the molecular mechanisms of cell death and a range of distinct types of cell death have been reported. Cell death can be classified into two major classes—programmed cell death and unprogrammed cell death (Figure 1). Mechanistically, programmed cell death is subclassified as apoptotic programmed, consisting of apoptosis, and non-apoptotic programmed, consisting of ferroptosis, pyroptosis, anoikis, NETosis, and necroptosis. The unprogrammed cell death on the other hand mainly refers to necrosis as the mechanism.

### 2.1. Apoptosis (Figure 2a)

Apoptosis, also called programmed cell death, is the result of an orderly cascade of a multistep enzymatic activity. Histologically, the characteristic features of apoptosis are (i) cell shrinkage, (ii) membrane blebbing, and (iii) condensation of the chromatin [17]. A molecular hallmark of apoptosis is DNA fragmentation, the nuclear genomic DNA is cleaved into ≈180 bp in length. Mechanistic studies reveal that apoptosis can be classified into two distinct pathways: the intrinsic pathway and extrinsic pathway. Both the intrinsic pathway and extrinsic pathway share caspase protease activation [18]. In the extrinsic pathway, apoptosis is initiated via death receptors, a group of cellular membrane proteins including tumor necrosis factor receptor (TNFR) and Fas/CD95. Once the legitimate ligands bind to the receptors on the surface of the cellular membrane, the receptors will be activated and downstream signaling will be culminated. By interacting with Fas-associated death domain protein (FADD) and other such adaptor proteins, the Death-inducing signaling complex (DISC) activates caspase 8, which then activates caspases 6, 7, and subsequently caspase 3, resulting finally in proteolysis of substrate and eventual cell death [19] (Figure 2a). The intrinsic pathway, also known as the mitochondrial pathway, is activated via responding to cellular stresses. Cellular stresses activate Bad/Bax, that release cytochrome C from the membrane of mitochondria to activate caspase 9, and caspase 9 cleaves caspase 3 to induce cell death [16] (Figure 2a).Apoptosis could be observed in physiological processes such as in embryonic development, and it extensively exists in the pathogenesis of various diseases as well.

### 2.2. Pyroptosis (Figure 2b)

Pyroptosis is a type of programmed cell death involving breakage of plasma-membrane, resulting in the release of proinflammatory intracellular contents [20]. Caspases 1, 4, 5, and 11 play an important role in pyroptosis [21]. Three different pathways for pyroptosis have been proposed so far(Figure 2b): the caspase 1 dependent pathway also called the canonical inflammasome pathway, the noncanonical inflammasome pathway consisting of caspases 4, 5, and 11, and most recently proposed the caspase 3 dependent pathway [21,22].In the canonical inflammasome pathway, intracellular pattern recognition receptors (PRR) receive signal stimulation in the form of pathogen-associated molecular patterns (PAMPs) and danger-associated molecular patterns (DAMPs). Subsequently, the inflammatory body is assembled and an intracellular macromolecular protein complex is produced, leading to the activation of caspase 1, which thereby causes secretion ofIL-1β and IL-18 on one hand, and cleavage of gasdermin-D (GSDMD) to generate two types of ends–reactive amino (N) and carboxyl (C). All this leads to the destruction of cellular structure and subsequent cell lysis and cell death [21].

In the noncanonical inflammasome pathway, caspase-11, derived from mice, and caspases 4 and 5, both derived from humans, can induce pyroptosis. In this pathway, binding of the caspase to LPS occurs, leading to the cleavage of GSDMD which promotes pyroptosis [21]. The third pathway, which is caspase-3 dependent, is similar to the canonical inflammasome pathway, with the only difference being that it involves the cleavage of gasdermin E (GSDME), instead of GSDMD. This cleavage leads to the cell structure destruction and eventual pyroptosis [21].

### 2.3. NETosis (Figure 2c)

The dynamic process of activation and release of neutrophil extracellular traps (NETs) which may result in cellular death is termed as NETosis [23]. NADPH-oxidase production of reactive oxygen species (ROS) is suggested to be the starting point of the pathway, leading to the activation of protein-arginine deiminase 4 (PAD4), an enzyme responsible for chromatin decondensation in the neutrophil nucleus. The subsequent rupture of the nuclear envelope occurs when myeloperoxidase (MPO) and neutrophil elastase (NE) enters the nucleus. Suicidal NETosis and vital NETosis are the two forms, with the key differences being in the stimuli, timing, and the end result (Figure 2c). NETosis is observed in many cases of infection, and it is also observed in no-infectious conditions.

### 2.4. Anoikis (Figure 2d)

Anoikis is a particular type of apoptosis occurring in the cells, either in absence of attachment to extracellular matrix (ECM) or the cells getting adhered to an inappropriate location [19]. Two apoptotic pathways are responsible for inducing the anoikis program, namely the intrinsic pathway, involving the perturbation of mitochondria, and the extrinsic pathway, consisting of cell surface death receptors getting triggered. The proteins of the Bcl-2 family are critical to both the pathways [19]. There are three categorizations of Bcl-2 proteins: anti-apoptotic proteins, multidomain pro-apoptotic proteins, and BH3-only proteins, which are also pro-apoptotic. Anoikis shares common downstream pathways of apoptosis (Figure 2d). In the intrinsic pathway, BH3-only proteins, including Bim, Hrk, Bmf, Bad, Bik, Puma, and Noxa, promote the activation of Bax/Bak. Among them, the direct promotion is done by Bid and Bim (activators), while the other members also known as sensitizers, promote indirectly by counteracting the anti-apoptotic functions of Bcl–2. The release of cytochrome c to the cytoplasm is the final step, leading to apoptosome formation, and thereby activation of executioner caspases [19]. The extrinsic pathway, which is the death receptor pathway, is initiated by members of TNFR, such as Fas and TNFR1, which leads to the apoptotic signal transductions.

### 2.5. Ferroptosis (Figure 2e)

Ferroptosis can be defined as a death program which is executed by selective oxidation of arachidonic acid-phosphatidylethanolamines (AA-PE) by 15-lipoxygenases [24]. It has been identified that the substrates and the products of this process are arachidonoyl-phosphatidylethanoamine (AA-PE) and 15-hydroperoxy-AA-PE (15-HOO-AA-PE), respectively [25,26]. Since lipid peroxide is an important participant in ferroptosis (Figure 2e), this categorizes two processes, one which promotes lipid peroxide formation and thereby ROS generation, and other which inhibits the reduction of lipid peroxides [27].

Inhibition of reduction of lipid peroxide is achieved either by deactivating GPX4 or by depleting its cofactor glutathione (GSH) [28,29].The inhibitors which directly or indirectly deactivate GPX4 are Erastin, RSL3, FIN56, and FINO2, while GSH depletion is done by Erastin alone [27]. On the other hand, the enzymes which promote the lipid peroxide formation are lysophosphatidylcholine acyltransferase 3 (LPCAT3), lipoxygenases (LOXs), and acyl-CoA synthetase long-chain family 4 (ACSL4) [25,30,31,32,33]. Additionally, production of ROS by fenton reaction and lipid autoxidation are also the pathways for ferroptosis [34,35].

### 2.6. Necroptosis (Figure 2f)

Necroptosis is a programmed form of necrosis [36]. Distinct with unregulated necrosis, cells can execute necrosis in a programmed manner independent of caspase activation. The tumor necrosis factor/tumor necrosis factor receptor (TNF/TNFR) signaling pathway has revealed to initiate necroptosis. Activation of TNFR signals TNFR-associated death protein (TRADD) and TNFR-associated factor 2 (TRAF2) that recruits Receptor-interacting protein kinase 1/ Receptor-interacting protein kinase 3 (RIPK1/RIPK3) to form a necrosome. The necrosome promotes mixed lineage kinase domain-like protein (MLKL) phosphorylation, allowing the MLKL to insert into and permeabilize plasma membranes and organelles that result in cell death (Figure 2f).

### 2.7. Necrosis (Figure 2g)

Necrosis is a form of cell death due to an external or internal injury that results in a premature death of the cell in the living tissues with autolysis. A variety of injury factors including microbial infection, toxins, or trauma induce necrosis by a digestion of the cell components. Necrosis is detrimental and can be fatal. Morphologically, necrosis is characterized by the nuclei of the cells fading away and shrinking and disruption of the membrane of the cells or organelle. It is in general believed that necrosis is an unregulated process (Figure 2g).

## 3. *P. aeruginosa* Induced Lung Epithelial Cell Death

### 3.1. P. aeruginosa Triggers Apoptosis in Lung Epithelial Cells (Figure 3a)

*P. aeruginosa* uses distinct mechanisms for initiating the cell death of the infected host cells [37]. The most studied cell death is apoptosis (Table 1). A well-studied paradigm is the extrinsic apoptotic pathway Fas (CD95)/Fas ligand signaling in lung epithelial cells. Following the infection of epithelial cells, either by in vitro or in vivo route, *P. aeruginosa* causes an upregulation of Fas/Fas ligand on the cell surface [38], an endogenous receptor ligand pair considered to be one of the most important ones responsible for the triggering of apoptosis. Bacteria derived type III secretion system (T3SS) upregulates the Fas/Fas ligand pair in *P. aeruginosa* infected cells and bacteria who do not possess a functional T3SS almost fail to cause apoptosis in epithelial cells. Upon upregulation, ligation of Fas by Fas ligand leads to induction of caspase 8 and caspase 3 activation, release of mitochondrial cytochrome C, as well asc-Jun N-terminal protein kinase(JNK) activation [37]. It is also found that reactive oxygen intermediates seem to be crucial for inducing *P. aeruginosa* triggered death [39]. Genetic studies using cells or mice genetically deficient for functional Fas or Fas ligand have shown the significance of the Fas/Fas ligand system for *P. aeruginosa* triggered cell death [38]. There was no response to *P. aeruginosa* infections resulting in induction of apoptosis in epithelial cells obtained in vivo from mice deficient in Fas or Fas ligand or ex vivo fibroblasts deficient in either Fas or Fas ligand. Apoptosis of lung epithelial cells upon getting infected with *P. aeruginosa* forms a crucial part of the host defense against infection from *P. aeruginosa*, which is contrary to the role of apoptosis due to infection from some other bacteria such as *Shigella flexneri* [37]. This was demonstrated by comparative in vivo pulmonary infection of Fas or Fas ligand deficient mice and normal mice. The former rapidly developed sepsis and died, while the latter completely cleared the infection within a period of few days. Another study has also demonstrated the protective effect of apoptosis triggered by bacteria [40]. The finding of that study is that a ced 3 and ced 4 regulated death of worm gonad cells is triggered when the worm *Caenorhabditis elegans (C. elegans)* is infected with *Salmonella Typhimurium* (*S. typhimurium)*. However, apoptosis inhibition in those cells caused by mutations of ced3 and ced4, respectively, leads to hyper-sensitization of *C. elegans* to the *S. typhimurium* infection [37].

*P. aeruginosa* may induce extrinsic apoptosis without ligand binding by directly polymerizing the death receptors within lipid rafts, that process activates the death receptor. A recent study reported that the quorum-sensing autoinducers N-(3-oxo-dodecanoyl) homoserine lactone in *P. aeruginosa* triggers significant cell death in B-lymphocytes, T-lymphocytes, dendritic cells, microphages, and monocytes. N-(3-oxo-dodecanoyl) homoserine lactone is able to incorporate into the host cell plasma membrane, plasma membrane lipid components containing cholesterol, sphingomyelin, and 1-2-dioleoyl-sn-glycero-3-phosphocholine (DOPC) efficiently retain 3-oc [42]. Addition of cholesterol inhibitor methyl-β-cyclodextrin dissolved cholesterol into the solution, thus disrupted the retention of 3-oc in the lipid domains. Retention of 3-oc in the plasma membrane dramatically changed the appearance of the presumed solid ordered lipid domains with a clear dissolution of the elevated plains of lipid domains to form smaller sized lipid rafts and collapse of the cholesterol and sphingomyelin-rich lipid domains, which was unrelated to membrane leakiness. However, domains formed with a mixture of DOPC and DPPG, unrelated to lipid rafts, were minimally affected. This event expels TNFR1 into the disordered lipid phase for its spontaneous trimerization without its ligand and drives Caspase 8-Caspase 3 mediated apoptosis [42].

*P. aeruginosa* activates cell surface receptors, thereby eliciting a cascade for intracellular signaling [43]. This leads to an increase in the communication at gap junction of the protein Cx43. It is observed that the expression of Cx43 displayed regulation in opposite directions exerted by JNK and p38 MAPKs. JNK inhibitor caused an increase in the PAO1-induced apoptosis, but the apoptosis was prevented by lentiviral expression of a Cx43-specific short hairpin RNA. Interestingly it is found that JNK activity undergoes upregulation by pharmacological restriction of CFTR in Calu-3 cells, however, when the CF airway cell line (CF15 cells) is corrected by adenoviral expression of CFTR, it causes a reduction in this MAPK being activated. This CTFR inhibition is also linked to the downregulation of Cx43 and thereby reduction in apoptosis. These results indicate that Cx43 expression forms a part of the response of airway epithelial cells by maintaining a balance of survival and apoptosis [43].

Another study demonstrated that *P. aeruginosa* induces the intrinsic apoptotic pathway via mitochondria. It has identified an important pathway responsible for the pathogenesis of lung injury caused by a *P. aeruginosa* linked virulence factor [41]. 3-oxo-C12-HSL is a principal quorum sensing molecule of *P. aeruginosa*, which causes disruption of mitochondrial morphology and promotes mitochondrial DNA oxidative injury. 3-oxo-C12-HSL downregulates the expression of peroxisome proliferator-activated receptor-γ coactivator-1α (PGC-1α), which is responsible for regulating biogenesis of mitochondria, acting as an antioxidant defense, as well as playing a role in cellular respiration. Overexpression of PGC-1α reduces the inhibition in cellular respiration produced due to 3-oxo-C12-HSL. It is also observed that the pharmacologic activation of PGC-1α causes restoration of barrier integrity in cells treated with 3-oxo-C12-HSL.

*P. aeruginosa* mediated apoptosis is stringently regulated in a few ways. Oral bacteria may be responsible for modulating the adhesion and invasion of respiratory pathogens to epithelial cells [53,54]. The presence of oral pathogenic bacteria leads to more *P. aeruginosa* infected epithelial cells via augmented mucosal surface invasion and lung epithelial cell apoptosis. Furthermore, oral as well as respiratory bacteria appear to induce the proinflammatory cytokines release from respiratory epithelial cell lines in vitro [54]. Oral bacteria may be responsible for altering the local microenvironment, thereby facilitating the onset and/or progression of respiratory disease in susceptible individuals [54].

It has been demonstrated that physiological levels of acidosis cause enhancement of epithelial cell cytotoxicity during a *P. aeruginosa* infection [55]. It has been shown in the study that the increase in epithelial cytotoxicity during acidosis is caused due to reduction in antimicrobial activity. Additionally, it has also been established that the epithelial-derived bactericidal activity is dependent on pH. Overall, these findings have provided key insights into the contribution of changes in extracellular pH to the pathogenesis of *P. aeruginosa* infections.

### 3.2. Epithelial Anoikis Caused by P. aeruginosa Infection (Figure 3b)

It has been reported that the GAP as well as the ADPRT domains of Exoenzyme T (ExoT) derived from *P. aeruginosa* contribute to ExoT-induced apoptosis in epithelial cells [50]. Crk adaptor protein gets transformed into a cytotoxin due to the ADPRT domain activity, inducing atypical anoikis apoptosis due to interference in the signaling of integrin survival [51]. However, the mechanism for the GAP-induced apoptosis was unknown until a more recent study shed light on this. It has been demonstrated in this study that the GAP domain of ExoT is responsible for triggering the mitochondrial intrinsic pathway of apoptosis [52]. The data from the study shows that intoxication of GAP leads to (i) Bax, Bid, and to a lesser extent Bim, getting activated and accumulated in the mitochondrial membrane; (ii) mitochondrial membrane losing its potential as well as cytochrome c release; (iii) activation of the initiator caspase-9 getting activated, resulting in the activation of the executioner caspase-3 thereby culminating in cell demise [52].

### 3.3. P. aeruginosa Induced Pyroptosis (Figure 3c)

Neutrophil pyroptosis is caused due to acute *P. aeruginosa* infection via inflammasome signaling. It has been reported that flagellin within *P. aeruginosa* and mitochondrial reactive oxygen species (ROS) cause neutrophil pyroptosis upon acute lung infection through CASP1-dependent signaling in Nox2 mice, resulting in enhanced lung inflammation and injury [44]. It has been demonstrated that PAO1-induced pyroptosis depends on NLRC4 and Toll-like receptor 5 (TLR5) in neutrophils. The authors suggest that mitochondrial ROS plays a role as a bridge between NOX2-mediated signaling and inflammasome activation in neutrophils, caused due to bacterial infection.

### 3.4. P. aeruginosa Associated NETosis (Figure 3d)

Type I interferons (IFNs) regulate neutrophil activity, and therefore act as the first line of anti-bacterial host defense [45,46]. Infections with bacteria, such as *P. aeruginosa*, are often linked to the increase in type I IFN signaling in lung epithelial cells [47]. It has been demonstrated that type I IFN-mediated activation of neutrophils in lungs causes increase in NETosis, leads to the prominent tissue damage, and supports biofilm formation by *P. aeruginosa* and its persistence in the lung [48]. The role of PH in mediation of NETosis is also an important aspect [49]. It shows that increasing pH in neutrophils caused stimulation in Nox activity and ROS production, both of which are essential agonist-induced Nox-dependent NETosis. At higher pH, neutrophil proteases, considered to be important for NETosis [50,51], could better cleave the histones after entering NETotic nuclei. Hence, high pH enables NETosis while low pH suppresses NETosis. It is also mentioned in the study that compounds such as sodium bicarbonate and THAM, which are clinically used, effectively increase pH and promote NETosis, which suggests the possible role of these compounds in correcting defective NETosis in vivo.

### 3.5. Epithelial Ferroptosis in Context of P. aeruginosa (Figure 3e)

In a new study, it has been reported that a mutant of *P. aeruginosa* producing biofilm induces ferroptosis in human bronchial epithelial (HBE) cells [24]. This is done by enhancing expression of pLoxA as well as oxidizing host cell AA-PE to 15-HOOAA-PE. It was observed that clinical *P. aeruginosa* isolates from patients suffering from prolonged lower respiratory infection resulted in pLoxA-dependent ferroptosis of HBE cells. High levels of 15-HOO-AA-PE in airway tissues from cystic fibrosis patients were detected using global redox phosphorlipidomics. However, in CF patients without *P. aeruginosa* in airway cultures, such elevated levels were not detected. Based on the assumption that disruption of epithelial barrier and immune-regulatory functions play a key role for pathogenesis of *P. aeruginosa*-related respiratory diseases, pLoxA as a potential new therapeutic target was proposed.

## 4. Cell Death Mediation and Therapeutic Strategies

### 4.1. Role of IL-15 in Apoptosis Prevention

Interleukin-15 (IL-15) belongs to the γ-chain family of cytokines, with IL-2, IL-4, IL-7, IL-9, and IL-21 being the other members [56,57,58,59,60]. IL-15 is unique among all of its family members on account of its pattern of receptor expression, which is observed in DC, NK, and CD8 T cells [61]. Due to this fact, IL-15 performs coordination of the response of these innate and adaptive immune cells for protecting the host [62,63,64,65,66]. IL-15 blocks sepsis-induced apoptosis in NK cells, dendritic cells, and CD8 T cells [61]. Sepsis induced gut epithelial apoptosis also decreases on account of IL-15. Furthermore IL-15 therapy increases antiapoptotic Bcl-2 and decreases proapoptotic Bim and PUMA. IL-15 increases circulating IFN-γ, along with the percentage of NK cells that produced IFN-γ. Lastly, IL-15 is responsible for increase in survival in both models of cecal ligation and puncture and *P. aeruginosa* pneumonia. Thus IL-15 is a potential novel therapy for the deadly *P. aeruginosa* caused disease.

### 4.2. Platelets Inhibit Epithelial Cell Apoptosis

A study has found that platelets are responsible for attenuating pathogen induced lung injury and protecting from lung epithelial apoptosis [7]. Platelet deficiency was identified to associate with severe disruption of alveolar-capillary barrier in live bacteria as well as bacterial exoproduct models representing *P. aeruginosa* infection. Platelets play a role in alveolar-capillary barrier homeostasis, but can also enter the airspace and release protective factors at the time of infection, thereby limiting alveolar epithelial cell death and reducing the further effect of lung injury. Lastly, the study also identified a potential protective role for platelet granule factors against lung injury triggered by pathogen as well as showed that platelet releasates are responsible for attenuating in vitro lung epithelial cell death as well as disrupting, independent of the whole platelets, the lung vascular barrier in mice deficient in platelets.

### 4.3. Mediation of Epithelial Cell Death in P. aeruginosa Pneumonia by Morf4l1

Mortality factor 4 like 1 (Morf4l1) is a protein playing a role in chromatin remodeling [67]. Its fundamentally low level of expression in the lung is on account of its short life via continuous ubiquitin proteasomal degradation, which is mediated by Fbxl18, an orphan ubiquitin E3 ligase subunit [67]. There is however an increase in the expression of Morf4l1 in humans with pneumonia and an upregulation in lung epithelia on getting exposed to *P. aeruginosa* or lipopolysaccharide (LPS) [67]. In a mouse model of pneumonia induced by *P. aeruginosa*, it was observed that Morf4l1 is stabilized due to acetylation that protects it from Fbxl18-mediated degradation. After the mice were infected with *P. aeruginosa*, overexpression of Morf4l1 enhanced lung epithelial cell death, in contrast to restored cell viability when it was depleted. A U.S. Food and Drug Administration-approved thrombin inhibitor argatroban, was identified to be a Morf411 antagonist using in-silico modeling as well as drug-target interaction studies. It was observed that Morf4l1-dependent histone acetylation, inhibited by argatroban, caused a reduction in its cytotoxicity and thereby improved the survival of mice with experimental lung injury at doses where the anticoagulant activity was none. This study uncovered a new biological mechanism, but also identified a potential molecular target for non-antibiotic pharmacological therapy in severe pulmonary infection.

### 4.4. Inhibition of Cellular Apoptosis and Autophagy Due to Tremella Polysaccharides

The endotoxin LPS derived from *P. aeruginosa* is able to induce apoptosis and autophagy, as well as increases the production of reactive oxygen species (ROS) in a time dependent manner in human epithelial A549 lung cancer cells [68]. It was also observed that LPS treatment suppressed sirtuin 1 (SIRT1) protein expression in the cells. The notable aspect of the study was the demonstration of SIRT1 activation due to *Tremella* polysaccharides activating SIRT1, resulting in an increase in the p62 expression, decrease of p53 acetylation as well as B-cell lymphoma 2-associated X protein expression, which subsequently attenuated LPS-induced apoptotic cell death and autophagy. These results open several potential avenues for treating the *P. aeruginosa* infection.

## 5. Conclusions and Future Directions

*P. aeruginosa* is a highly versatile microorganism, which continues to astonish us by evolving and acquiring new and unexplored modes of niche adaptation, lifestyle, as well as pathogenicity. A major mechanism of the pathogenicity of *P. aeruginosa* is causing host cell death. Among distinct cell death pathways that *P. aeruginosa* elicited, apoptosis is well studied. It may use a range of mechanisms to cause host cell apoptosis both in intrinsic and extrinsic apoptotic pathways. Meanwhile, a few studies focused on the regulation of *P. aeruginosa* induced apoptosis. A number of cell death pathways other than apoptosis have been reported (tableure 3). The importance of each pathway should be weighted in the pathogenesis of cell death by *P. aeruginosa*. Several attempts have been thrown into targeting *P. aeruginosa* induced cell death, these include IL–15, platelet factors, chromatin modulator Morf4l1, or LPS antagonists. Continued focus needs to be given on understanding the mechanisms of cell death that may lead to the identification of novel targeting therapeutics. Preclinical trials are required to verify the efficacy of the discovered targets. *P. aeruginosa* is the leading pathogen isolated in patients within critical care units. Secondary infection with *P. aeruginosa* worsens the prognosis of the patients. In the current pandemic of COVID-19, it is estimated that approximately 5% of the total patients are treated within critical care units [69,70]. Among these patients, sepsis has been noticed as a major contributor to the poor prognosis. The importance of the secondary infection with *P. aeruginosa* in the severe COVID-19 patients with long-term ventilation has yet to be understood. Prevention of the nosocomial *P. aeruginosa* infection may be critical to reduce the mortality of COVID-19.

## Figures and Tables

**Figure 1 ijms-21-05356-f001:**
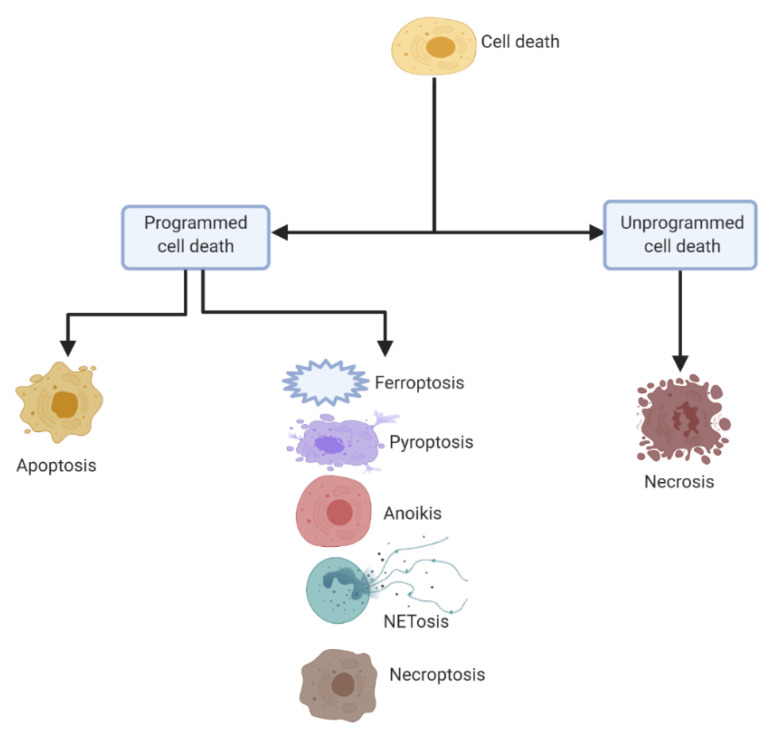
General classification of cell death. The broad classification is programmed and unprogrammed. The programmed cell death is subclassified as apoptotic and non-apoptotic, with apoptotic consisting of apoptosis, while non-apoptotic consisting of ferroptosis, pyroptosis, anoikis, NETosis, and necroptosis. Unprogrammed cell death has necrosis as the mechanism (all figures were created using Biorender.com).

**Figure 2 ijms-21-05356-f002:**
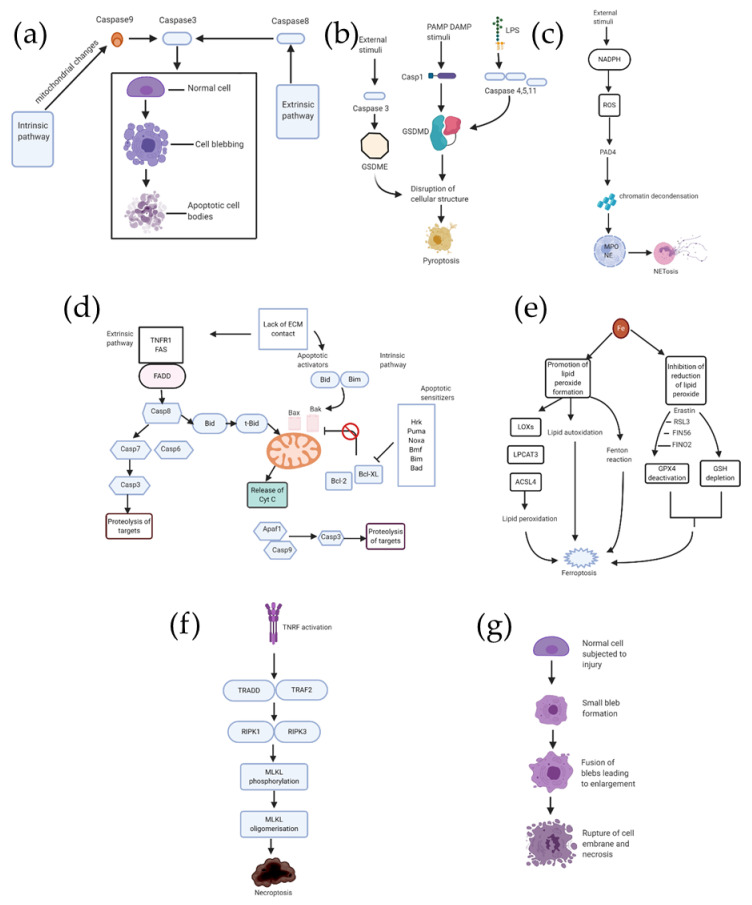
Schema for the mechanisms for different types of cell deaths. Part (**a**) displays two important pathways for apoptosis, namely intrinsic and extrinsic pathways, both converging at caspase-3 as the executioner pathway. Part (**b**) shows the three underlying pathways for pyroptosis involving caspases 1,3,4,5, and 11. Part (**c**) is the schematic representation for NETosis. Part (**d**) exhibits the two pathways for anoikis, intrinsic and extrinsic, respectively. Part (**e**) outlines the mechanistic pathways associated with ferroptosis, promotion of lipid peroxide formation being one path and inhibition of reduction of lipid peroxide being the other. TNF/TNFR signaling pathway for necroptosis via MLKL phosphorylation is displayed in part (**f**). Finally, part (**g**) represents the different stages in a general necrosis mechanism.

**Figure 3 ijms-21-05356-f003:**
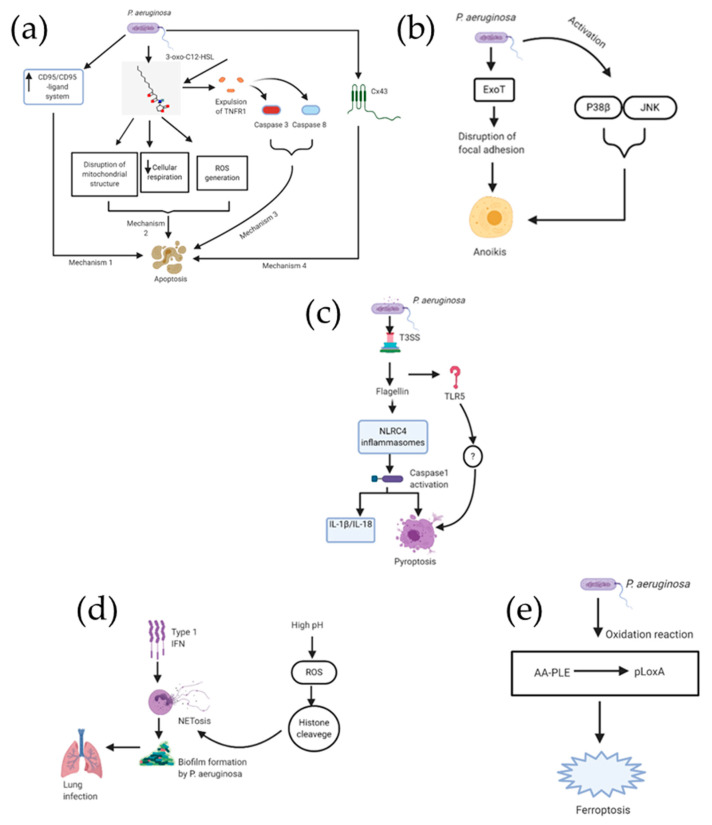
Schematic representation for the different cell death mechanisms caused due to *P. aeruginosa* infection. Part (**a**) consists of four important pathways for apoptosis, part (**b**) displays ExoT induced as well as activated P38-JNK induced anoikis, part (**c**) focuses on the two pathways associated with pyroptosis, part (**d**) is a cartoon representation of NEtosis pathway. Finally, part (**e**) is a single mechanism for *P. aeruginosa* induced ferroptosis involving peroxide formation and reactive oxygen species (ROS) generation, since ferroptosis is a newer cell death mechanism still being investigated for additional pathways.

**Table 1 ijms-21-05356-t001:** A summary of different mechanisms underlying the different types of epithelial cell death.

Type of Cell Death	Description	Mechanism	Reference Numbers
Apoptosis	CD95/CD95-ligand system for *P. aeruginosa* triggered apoptosis	*P. aeruginosa* causes an up-regulation of CD95/CD95ligand on the cell surface, responsible for the triggering of apoptosis	[37,38,39,40]
Apoptosis	Disruption of mitochondrial morphology using 3-oxo-C12-HSL	Quorum sensing molecule 3-oxo-C12-HSL activates the apoptosis by disrupting the mitochondrial structure, attenuating cellular respiration and inducing ROS generation	[41]
Apoptosis	Caspase 3-caspase 8-mediated apoptosis	Necrosis factor receptor 1 expelled into the disordered lipid phase triggers cell death	[42]
Apoptosis	Apoptosis due to Cx43-mediated cell-to-cell communication	Cx43-mediated gap junctional communication enhances apoptosis in PAO1-infected airway epithelial cells, while on the other hand JNK signaling inhibits Cx43 function	[43]
Pyroptosis	PAO1 flagellin induced CASP1-dependent neutrophil pyroptosis	PAO1-induced pyroptosis depends on NLRC4 and Toll-like receptor 5 (TLR5) in neutrophils	[44]
NETosis	Type I interferon associated NETosis	Excessive activation of neutrophils by type I IFNs causes aboost in NETosis which triggers biofilm formation by *P. aeruginosa,* thereby supporting its persistence in the infected lung.	[45,46,47,48]
NETosis	NADPH Oxidase-Dependent NETosis	Increase in pH in neutrophils stimulates Nox activity and ROS production requiredl for NETosis	[49]
Anoikis	*Pseudomonas aeruginosa* ExoT induced atypical anoikis	GAP domain of ExoT is responsible for triggering the mitochondrial intrinsic pathway of anoikis apoptosis	[50,51,52]
Ferroptosis	*P. aeruginosa* produced biofilm induces ferroptosis	Caused by enhancing expression of pLoxA as well as oxidising host cell AA-PE to 15-HOOAA-PE	[24]

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
