# Peer review of "Pseudomonas Aeruginosa* Induced Cell Death in Acute Lung Injury and Acute Respiratory Distress Syndrome"

_ijms, 2020, doi:10.3390/ijms21155356_

Round 1
Reviewer 1 Report
The authors reviewed the main clinical benefits of mechanisms of Pseudomonas aeruginosa induced cell death in acute lung Injury. The article is well presented and written, and the figures are well designed. However, I believe that the authors could make some changes to improve the clarity of the manuscript.
- Some reference sequence is wrong. For example, more than 10 million patients have been diagnosed with COVID-19 with half of million death, and the numbers of confirmed cases and death are continuously increasing (70). This should be reference 1.
- The authors did not cite references in many important descriptions. For example, the authors mentioned “P. aeruginosa is almost 100% positive in the respiratory tract of patient with long-term ventilation more than 7 days”. This number, 100% is unreasonable, please list references
- The authors mentioned the importance and impact of COVID 19. Is there any reference describing the relationship between COVID and Pseudomonas aeruginosa?
- Pseudomonas aeruginosa causes different conditions of patients, such as colonization, pneumonia without acute lung injury, pneumonia with acute lung injury. What is the different mechanism in this part?
- The authors described many important mechanisms and drew many concise and clear figures. It will be clearer to make a table to summarize these mechanisms.
Author Response
1. Some reference sequence is wrong. For example, more than 10 million patients have been diagnosed with COVID-19 with half of million death, and the numbers of confirmed cases and death are continuously increasing (70). This should be reference 1
Response: We have corrected the reference sequence as pointed out.
2. The authors did not cite references in many important descriptions. For example, the authors mentioned “P. aeruginosa is almost 100% positive in the respiratory tract of patient with long-term ventilation more than 7 days”. This number, 100% is unreasonable, please list references
Response: We have added the reference for the mentioned statement and have also modified it from “almost 100% patients” to “many of the patients”.
3. The authors mentioned the importance and impact of COVID 19. Is there any reference describing the relationship between COVID and Pseudomonas aeruginosa?
Response: COVID-19 has been mentioned in the introduction of the paper in context of the current pandemic situation as well as it causing ALI/ARDS, which is relevant to the paper. However, the exact relationship between COVID and Pseudomonas aeruginosais still unknown considering the disease is really new, and therefore can be a basis for potential future study.
4. Pseudomonas aeruginosa causes different conditions of patients, such as colonization, pneumonia without acute lung injury, pneumonia with acute lung injury. What is the different mechanism in this part?
Response: The mechanism for Pseudomonas aeruginosa colonization is by JNK activationpathway which is described in line 212 and 245-248 as well as included in Fig 3(d). With regards to the lung injury in pneumonia, as per a review article in 2017 (Lee, Kyung-Yil. “Pneumonia, Acute Respiratory Distress Syndrome, and Early Immune-Modulator Therapy.” International journal of molecular sciences vol. 18,2 388. 11 Feb. 2017, doi:10.3390/ijms18020388), the exact mechanism is still unknown.
5. The authors described many important mechanisms and drew many concise and clear figures. It will be clearer to make a table to summarize these mechanisms.
Response: The table is already present on page 17 of the manuscript, containing a summary of the mechanisms along with the reference numbers.
Reviewer 2 Report
The authors have given an excellent review of different types of cell death caused by P. aeruginosa, which lead to acute lung injury and acute respiratory distress syndrome, and their potential therapeutics. Overall, this article is well written and provide us with a complete understanding of the mechanism of P. aeruginosa –induced cell death. There are only a few points needed to be attended.
- Page 1, line 40: The sentence, “ aeruginosa is almost 100% positive in the respiratory tract of patient with long-term ventilation more than 7 days,” seems not always the case in different regions of the world. The sentence does not indicate where it is quoted from.
- Page 6, line 188: Should the sentence “Necroptosis is a programmed form of necroptosis” be modified to “Necroptosis is a programmed form of necrosis”?
- Please proofread the entire article carefully, because there are some words misspelling and no full spelling in front of abbreviation, such as: page 7, lines 234-236: I am not sure what the full name spelling of 3-oc is; page 7, line 238: “lipid tafts” should be “lipid rafts”; page 8, line 304, “inNox” should be “in Nox”.
- The statements of each type of aeruginosa-induced cell death should have the indicated number of the figures they correspond to.
Author Response
1. Page 1, line 40: The sentence, “aeruginosa is almost 100% positive in the respiratory tract of patient with long-term ventilation more than 7 days,” seems not always the case in different regions of the world. The sentence does not indicate where it is quoted from.
Response: We thank the reviewer for the suggestive commentaries and we have modified the sentence and provided reference as stated earlier in response to the second point of the first reviewer.
2. Page 6, line 188: Should the sentence “Necroptosis is a programmed form of necroptosis” be modified to “Necroptosis is a programmed form of necrosis”?
Response: We have edited the sentence to reflect the mentioned correction.
3. Please proofread the entire article carefully, because there are some words misspelling and no full spelling in front of abbreviation, such as: page 7, lines 234-236: I am not sure what the full name spelling of 3-oc is; page 7, line 238: “lipid tafts” should be “lipid rafts”; page 8, line 304, “inNox” should be “in Nox”.
Response: We have made the suggested corrections. 3-oc stands for (3-oxo-dodecanoyl) homoserine lactone, which we have added in the abbreviation list as the last abbreviation.
4. The statements of each type of aeruginosa-induced cell death should have the indicated number of the figures they correspond to.
Response: We have added the figure numbers in the statements of respective cell death types.